# Cytokine Adjuvants IL-7 and IL-15 Improve Humoral Responses of a SHIV LentiDNA Vaccine in Animal Models

**DOI:** 10.3390/vaccines10030461

**Published:** 2022-03-17

**Authors:** Laury-Anne Leroy, Alice Mac Donald, Aditi Kandlur, Deepanwita Bose, Peng Xiao, Jean Gagnon, François Villinger, Yahia Chebloune

**Affiliations:** 1PAVAL Lab., Institut National de Recherche d’Agriculture et Environnement, Université Grenoble Alpes, 38400 Saint Martin d’Hères, France; laury-anne.leroy@univ-grenoble-alpes.fr (L.-A.L.); alice.mac.donald@umontreal.ca (A.M.D.); admeem@gmail.com (A.K.); jgagnon@fastmail.com (J.G.); 2New Iberia Research Center, University of Louisiana at Lafayette, New Iberia, LA 70560, USA; deepanwita.bose@louisiana.edu (D.B.); peng.xiao@louisiana.edu (P.X.); francois.villinger@louisiana.edu (F.V.)

**Keywords:** HIV-1, DNA vaccine, CAEV LTR, LentiDNA, SHIV, IL-7, IL-15, *Rhesus macaques*

## Abstract

HIV-1 remains a major public health issue worldwide in spite of efficacious antiviral therapies, but with no cure or preventive vaccine. The latter has been very challenging, as virus infection is associated with numerous escape mechanisms from host specific immunity and the correlates of protection remain incompletely understood. We have developed an innovative vaccine strategy, inspired by the efficacy of live-attenuated virus, but with the safety of a DNA vaccine, to confer both cellular and humoral responses. The CAL-SHIV-IN^−^ lentiDNA vaccine comprises the backbone of the pathogenic SHIV_KU2_ genome, able to mimic the early phase of viral infection, but with a deleted integrase gene to ensure safety precluding integration within the host genome. This vaccine prototype, constitutively expressing viral antigen under the CAEV LTR promoter, elicited a variety of vaccine-specific, persistent CD4 and CD8 T cells against SIV-Gag and Nef up to 80 weeks post-immunization in cynomolgus macaques. Furthermore, these specific responses led to antiviral control of the pathogenic SIV_mac251_. To further improve the efficacy of this vaccine, we incorporated the IL-7 or IL-15 genes into the CAL-SHIV-IN^−^ plasmid DNA in efforts to increase the pool of vaccine-specific memory T cells. In this study, we examined the immunogenicity of the two co-injected lentiDNA vaccines CAL-SHIV-IN^−^ IRES IL-7 and CAL-SHIV-IN^−^ IRES IL-15 in BALB/cJ mice and rhesus macaques and compared the immune responses with those generated by the parental vaccine CAL-SHIV-IN^−^. This co-immunization elicited potent vaccine-specific CD4 and CD8 T cells both in mice and rhesus macaques. Antibody-dependent cell-mediated cytotoxicity (ADCC) antibodies were detected up to 40 weeks post-immunization in both plasma and mucosal compartments of rhesus macaques and were enhanced by the cytokines.

## 1. Introduction

Since its discovery in the early 1980s, the zoonotic human immunodeficiency virus (HIV)-1 has been responsible for a long-lasting pandemic. The virus is mainly transmitted from human to human via sexual intercourse, or by contact with infected body fluids, e.g., via shared contaminated needles among intravenous drug users. Although the life cycle and the pathogenicity of this virus are now well-understood, lasting challenges remain to elucidate the correlates of protection that need to be elicited by a protective vaccine and how to eradicate the virus from infected patients on antiretroviral therapy (ART). HIV-1 infection rapidly seeds persistent reservoirs in various lymphoid organs as well as in sanctuary sites such as brain, testis, and gut-associated lymphoid tissues (GALT) [1] and in long-lived non-activated memory T cells, which outlive even long-term effective ART [2].

Many vaccine prototypes and strategies have already been tested against HIV, with only a few reaching phase III clinical trials, but not beyond [3]. From viral vectors expressing HIV-1 genes (HVTN 502, Merck trial) to recombinant envelope subunits to generate neutralizing antibodies (AIDSVAX trial), the combination of the two in the RV144 trial or the latest Uhambo or HVTN 702 trial that was stopped in 2020 due to absence of protection, none of these vaccines demonstrated robust lasting efficacy against HIV infection [4,5]. Among primate lentiviruses, live-attenuated simian immunodeficiency virus (SIV) is almost the only one able to elicit potent and durable immunogenicity and protection from rechallenges in non-human primates (NHP) [6]. However, this approach has also been demonstrated to retain significant pathogenicity and the ability to compensate for their genetic attenuation in vivo [7,8,9]. While this approach has been abandoned clinically, it has provided a model from which protective correlates could be identified [10,11]. Attempts to harness the power of virus inoculation using a variety of recombinant viral vectors (Ad5 [12], MVA [13], CMV [14]) have produced immune responses, but to date, limited data on protection. In addition, the use of such vectors is generally limited by vector-specific immune responses induced by natural virus infection or primed by the vaccine, precluding booster immunizations with the same vector. This was illustrated in the phase IIb trial “STEP” led by Merck. The vaccine was based on recombinant Ad5 viral vector expressing subtype B HIV-1 genes coding for Gag, Pol or Nef [15]. This trial was prematurely stopped after the vaccinated cohort showed higher levels of infection. This was possibly caused by reactivation of Ad5-specific CD4 T cells that became the major HIV target. Indeed, around 90% of the tested population showed high Ad5 antibody levels [15]. In contrast, plasmid DNA immunization does not lead to responses against the vector, allowing for repeated injections of genetic material, but this approach has shown limited immunogenicity in NHPs and humans [16,17]. However, when strategies to improve immunogenicity (prime boost, electroporation, optimized gene expression, etc.) were applied, DNA vaccine injections led to Th1 cellular as well as humoral responses. Both types of immune responses are thought to be necessary to fully control this chronic disease. Vaccines eliciting T-cell responses are becoming an important objective, as many studies on macaques and humans showed that cytotoxic CD8 T cells correlate with protection against HIV-1 [18,19]. DNA vaccines have been markedly enhanced by incorporation of electroporation (EP) as a method to enhance DNA uptake by up to 10- to 100-fold [20] into cells while also causing low-level inflammation. Administered intramuscularly, EP promotes the release of cytokines such as MIP-1β and recruitment of immune cells such as dendritic cells (DC), neutrophils, monocytes, B cells, and CD4 and CD8 T cells into tissues [21]. These cytokines and infiltrating cells do not damage tissues but rather contribute to immune priming [22]. Skin EP and particularly the intradermal route is also often utilized, as this compartment is rich in professional antigen-presenting cells (APCs) such as Langerhans cells that are able to cross-present vaccine antigens to naïve cells in draining lymph nodes. DNA immunization with EP generates both CD4 and CD8 T-cell responses.

In keeping with the unique ability of live attenuated vaccine to induce protection, we have developed an innovative lentiDNA vaccine, termed CAL-SHIV-IN^−^, that is able to perform a single cycle of replication in host cells. This strategy allowed for the release of viral immunogens and a reinforced presentation of the different antigens, leading to a robust immune response in a setting of prime boost strategy [23]. A single injection of this lentiviral vaccine prototype in cynomolgus macaques induced persistent SHIV-specific CD4 and CD8 T cells for up to 80 weeks post-immunization, with central memory (Tcm) and effector memory (Tem) cells secreting cytokines against Gag and Nef. More importantly, these responses allowed the animals to control viremia after repeated intrarectal challenges with SIV_mac251_ [24]. This vaccine was, however, not expected to prevent mucosal virus acquisition due to the mismatched envelopes, leaving ample room for improvement.

IL-7 and IL-15 are pleiotropic cytokines, known to increase proliferation, function, differentiation and survival of T cells [25,26,27]. Mice administered with a plasmid DNA vaccine formulation containing HIV-1 Gag with IL-7 or IL-15 (DermaVir) significantly enhanced Gag-specific central memory and effector memory CD8^+^ T cells [28,29]. Furthermore, IL-7 administered in the genital tract of female macaques vaccinated with Diphtheria/Tetanus (DT) led to the recruitment of mDCs, macrophages, NKs, and B and T cells to the lamina propria and APCs to epithelium. This recruitment was associated with a chemokine overexpression and faster, durable mucosal antibody responses in vaginal mucosa compared to DT immunization alone. [28,29]. Additionally, several clinical trials have incorporated IL-7 and IL-15 as vaccine adjuvants [28,29,30,31], cancer treatments [32], and strategies to counter the diminution or exhaustion of T cells in HIV-1 infection [33]. Therefore, we leveraged the adjuvant activity of these cytokines by insertion of cytokine-coding DNA cassettes under IRES control. The resulting lentiviral plasmids were then tested *in cellulo* for antigen expression and in vivo for their ability to induce antiviral responses. Results of these studies form the basis of this manuscript.

## 2. Materials and Methods

### 2.1. DNA Vaccine Construction

To generate CAL-SHIV-IN^−^ IRES-cytokine plasmids, we subcloned IRES-driving cytokine cassettes from expression plasmids for rhesus monkey cytokine (pNDgms-rmIL-15 and pNDgms-rmIL-7) provided by Dr François Villinger, New Iberia Research Center, LA (New Iberia, LA, USA). PCR products of cytokine sequences were ligated into the pGEM-T easy plasmid (Promega, Charbonnières-les-Bains, France). The GFP gene was removed from the pGEM-T IRES-GFP and replaced by the cytokine genes isolated from pGEM-T cytokines to generate pGEM-T IRES-cytokine plasmids. Insertion of IRES-cytokines in the genome of CAL-SHIV-IN^−^ followed the same Age I insertion site as for the CAL-SHIV-IN^−^ IRES-GFP vector, previously constructed in our laboratory and shown to efficiently express viral proteins and GFP (data not shown). Constructs having the IRES-cytokine inserted in the sense and antisense transcriptional orientations were isolated.

The selected constructs (Figure A1) were introduced into JM109 bacteria (Promega, Charbonnières-les-Bains, France) and cultured in LB medium supplemented with 100 μg/mL of ampicilin or 50 μg/mL of kanamycin for 20 h at 32 °C under agitation at 150 rpm. Briefly, 5 μL of ligation products were added to 50 μL of bacterial suspension, incubated on ice (15 min), followed by heat shock (90 s at 42 °C) and chilled on ice. Bacteria were then supplemented with 500 μL of SOB media (2% Bacto-Tryptone, 0.5% Bacto-Yeast extract, 10 mM NaCl, 2.5 mM KCl, 10 mM MgCl2) and incubated for 2 h at 37 °C, 200 rpm. Bacteria were plated on the surface of agar plates, supplemented with ampicillin or kanamycin, and then incubated at 32 °C overnight. Isolated single colonies were transferred into LB/ampicillin or LB/kanamycin medium and grown individually for miniprep screenings. The final vector was transformed into Stbl3 E. coli competent bacteria (ThermoFisher, Illkirch, France) and stored in a 50% glycerol stock at −80 °C. DNA extraction was performed using Nucleospin Plasmid Easy Pure kit or NucleoBond Xtra Maxi Endotoxin Free kit (Macherey-Nagel, Hœrdt, France), according to manufacturer’s instructions. The quality of extracted DNA was tested using agarose gel electrophoresis and quantification by measuring OD at 230, 260 and 280 nm with a NanoDrop UV apparatus. Deep sequencing (next-generation sequencing) was performed to verify the complete integrity of constructs.

### 2.2. Cell Culture

Human Lenti-X™ 293T cell line (Takara bio, San Jose, CA, USA) subcloned for high transfectability and high-titer virus production as well as HEK 293T were cultured in Dulbecco’s modified eagle medium (DMEM) (ThermoFisher, Illkirch, France) supplemented with 10% decomplemented fetal bovine serum (Eurobio, Les Ulis, France) and penicillin/streptomycin (Eurobio, Les Ulis, France), final concentration of 0.05 U/mL and 0.05 µg/mL, respectively (complete DMEM), and used for transfection assays. TZM-bl cells in complete DMEM were used for the neutralization assay. M8166 CD4 T lymphocytes permissive to SHIV infection were cultured in complete RPMI and used to detect cytopathic effects. CEM.NKr-CCR5 CD4^+^ target T cells maintained in complete RPMI medium and KHYG-1 NK cells maintained in complete RPMI medium supplemented with IL-2 (5 U/mL) and cyclosporine A (1 µg/mL) (Sigma-Aldrich, St. Louis, MO, USA) were used for the ADCC assay. Cell lines were kindly provided by the NIH AIDS Reagent Program, Division of AIDS, NIAID, NIH.

### 2.3. Expression of Viral Proteins and Cytokines IL-7 and IL-15

#### 2.3.1. Cell Transfection

HEK 293T cells were transfected with 5 µg DNA of recombinant expression plasmids of the cytokines pNDgms-rmIL-7 or pNDgms-rmIL-15 using the classical phosphate calcium method. Human Lenti-X™ 293T were transfected with DNA constructs CAL-SHIV-IN^−^, CAL-SHIV-IN^−^ IRES rmIL-7 or CAL-SHIV-IN^−^ IRES rmIL-15 using jetPEI^®^ Polyethylenimine reagent (Polyplus, Illkirch, France) at a DNA:jetPEI ratio of 1:3. Briefly, 1 × 10^6^ cells in suspension were seeded in a 6-well plate (VWR, Rosny-sous-Bois, France). Three µg of sterile, endotoxin-free DNA (>90% supercoiled) were complexed with jetPEI and added to the cell monolayer. Supernatants were collected at 2, 3 and 5 days post-transfection, cleared by centrifugation and frozen at −80 °C prior to ELISA.

#### 2.3.2. E Enzyme-Linked Immunosorbent Assay: ELISA

To assess viral protein expression, supernatants from transfected cells were quantified by the SIV Gag p27 kit (ABL, Rockville, MD, USA) according to the manufacturer’s recommendations. A linear regression curve-fit was created to determine the Gag p27 concentration. Supernatants from transfected cells were also tested for cytokine expression using the ELISA Human IL-7 or IL-15 DuoSet (R&D systems, Lille, France) kits according to the manufacturer’s instructions. Absorbance was measured at 450 nm with a POLARstar Omega Plate Reader Spectrophotometer (BMG Labtech, Champigny-sur-Marne, France), and blank was subtracted. Quantitative analysis of samples was performed using a four parameter logistic (4PL) curve-fit via MARS Data Analysis Software (BMG Labtech, Champigny-sur-Marne, France).

#### 2.3.3. Functional Analysis of Cytokines

To evaluate cytokine functions, we used cryopreserved human PBMCs from a BCG-vaccinated blood donor in a PHPC assay setting [23]. Briefly, PBMCs were thawed and incubated 24 h prior to use in the assay. Cells (2.5 × 10^5^) were seeded in an IFN-γ ELISpot assay (Mabtech, Sophia Antipolis, France) precoated plate and stimulated with 4 µg/mL of heat-inactivated and sonicated Polish BCG vaccine (Sanofi Pasteur MSD–50 µg/0.1 mL or 1.5-6 × 10^5^ viable units of Moreau, Brazil strain). Another fraction of PBMC (10^6^ cells/well) was cultured for 12 days to evaluate the precursor with high proliferation capacity (PHPC) memory cells in the presence of 4 µg/mL of the same antigen and stimulated with homeostatic cytokines. On day 3, cultures were supplemented with recombinant simian IL-2 (10 U/mL from the Resource for NHP immune Reagents) only. On day 7, a cocktail containing simian IL-2 (10 U/mL), recombinant simian IL-7 (Cytheris; now Revimmune Inc., Paris, France) and IL-15 or cytokine released in the culture medium of HEK 293T cells transfected with CAL-SHIV-IN^−^ IRES cytokines was added to the cultures. Recombinant and HEK-293T cell-produced IL-7 and IL-15 were all used at 100 pg/mL. On day 11, 2 × 10^5^ cells per well of expanded PBMCs were tested for BCG-specific responses by IFN-γ ELISpot assay.

### 2.4. Evaluation of Immunogenicity in Animal Models

#### 2.4.1. Ethical Statements

Animal studies were performed according to NIRC standard operating procedures (with the exception of noted exemptions in the study protocol) and according to the “Guide” of Animal Use in Research. All animal study protocols were reviewed and approved by the University of Louisiana at Lafayette Institutional Animal Care and Use Committee (IACUC) under approval 2019-8743-030.

Mice were housed in compliance with the “Guide”, the AWA and AWR. Five mice were housed per cage to allow social interactions, with enrichment such as cellulose, pieces of wood or cardboard rolls. Housing of mice was under controlled conditions of humidity, temperature, and light (12 h white light/12 h red light). Water and dry pellets were provided ad libitum. Mice were anesthetized with isoflurane vapor (Patterson Veterinary) and monitored until awake.

Macaques were housed in adjoining individual cages allowing social interactions, under controlled conditions of humidity, temperature, and light (12 h light/12 h dark cycles). They were fed 8773 Harlan Teklad NIB Diet Monkey Chow, or its equivalent, in amounts appropriate for the size of each animal. Water was provided ad libitum via automatic watering devices. Macaques were provided with environmental enrichment including toys and novel food items such as fruits or nuts. Food was withheld the morning of scheduled sedations to insure a minimum of 4 h fast prior to sedation. All collections and immunizations were performed under sedation with ketamine HCl, 5–10 mg/kg, and/or Telazol, 4–8 mg/kg.

#### 2.4.2. Mice

Groups of 4–6 week-old male and female BALB/cJ mice (Jackson Laboratory, Bay Harbor, ME, USA) were used and distributed as follows: 2 groups of 20 mice, group 1: injected with a total of 100 μg CAL-SHIV-IN^−^ in 100 μL sterile endotoxin-free PBS (Quality Biological); group 2: co-immunized with a total of 100 μg of a 1:1 mixture of CAL-SHIV-IN^−^ IRES rmIL-7 and CAL-SHIV-IN^−^ IRES rmIL-15, in 50 μL each of sterile, endotoxin-free PBS. Two mice in group 3 were used as negative control (non-immunized). Six weeks post-immunization, group 1 and 2 were boosted with the exact same procedure as the first immunization (Figure A2).

Briefly, mice were anesthetized under isoflurane vapors and shaved prior to immunization. A total of 100 µg of DNA was injected, 50 µg intradermally (ID) in the lower back of the mouse, and 50 µg by intramuscular (IM) route in the right-side quadriceps. Intradermal injection was immediately followed by electroporation using a CUY21 edit II electroporator, (BEX CO, LTD., Nagoya, Japan) and 10 mm diameter platinum tweezer electrodes (BTX, Holliston, MA, USA). To improve conductivity between skin and electrodes, a small amount of conductive gel (SignaGel, Parker, Fairfield, NJ, USA) was used. Impedance between the electrodes and the skin was checked before each electroporation. The following parameters were used: a constant square-wave current, with 1 pulse allowing creation of pores in the cell membrane (poration pulse Pp)–700 V/cm for 10 ms, and 5 pulses to drive the DNA into the cells (driving pulse Pd)–100 V/cm for 10 ms, spaced by 50 ms. A limit of 300 mA maximum was delivered (PdA). Interspace between the two electrodes was set to 0.2 cm, and the generator programmed on Pp = 140 V and Pd = 20 V.

Five mice per group and per time point (week 2, 4, 8 and 10 post-immunization) were euthanized to collect blood by intracardiac puncture and the spleen to characterize both humoral responses in serum and cellular responses in splenocytes.

#### 2.4.3. Macaques

Six male rhesus macaques (Macaca mulatta) from the New Iberia Research Center (NIRC) colony were used for this study and equally split into 2 groups according to body weight, age and MHC alleles (see Table A1). These animals were previously genotyped using DNA isolated with the QIA amp DNA Blood kit (Qiagen) and PCR with specific primers, for the evaluation of the MHC alleles Mamu A×001, B×008 and B×017, known to be factors favoring the virus control, and to distribute these profiles among groups.

The first group received the CAL-SHIV-IN^−^ vaccine, while the second group received a combination of the two DNA CAL-SHIV-IN^−^ IRES rmIL-7 and CAL-SHIV-IN^−^ IRES rmIL-15, mixed in equal amounts. Large-scale DNA vaccine preparations were resuspended aseptically in endotoxin-free PBS at a final concentration of 2 mg/mL. Each monkey received 5 mg of DNA vaccine for the prime injection: 4 mg were given by intramuscular injection (IM) and 1 mg intradermally (ID) in the scapular area. IM injections were administered in the 2 quadriceps (2 mg of DNA in each muscle), followed by electroporation (EP) (Cellectra 2000, Inovio Pharmaceuticals) with 3 pulses of 52 ms, 0.5 A, and 1 s between pulses. ID injections were separated in 2 sites with 0.5 mg/site, followed by electroporation (T820 Electro square wave porator BTX, Holliston, MA, USA) and 10 mm diameter platinum tweezer electrodes (BTX, Holliston, MA, USA) together with a small amount of conductive gel (SignaGel, Parker, Fairfield, NJ, USA). Six pulses of 10 ms, 110 V, with 1 s interval between pulses were applied. The same scheme was repeated for the homologous boost 16 weeks after the prime immunization.

Blood collection for PBMCs and plasma isolation: Blood samples were collected by venipuncture in EDTA vacutainer-type tubes from all animals and kept at room temperature until processing. PBMCs were isolated on Ficoll-Hypaque gradient using a standard method and used for the different assays.

Lymph node fine needle aspirate (FNA): Lymph node aspirates were collected from superficial left and/or right draining lymph nodes. The skin above the node was disinfected and the node entered with a 21–23 G needle. Once the needle was in the node, cells were aspirated, the syringe needle removed, and cells expelled into a tube containing complete RPMI medium. The tubes were transported on wet ice for processing.

Rectal Fluids: Two weck-cel sponges extended by a transfer pipet, pre-moistened with 50 μL of physiological saline, were introduced into the rectum inside a cut 5 mL pipet and pushed against the mucosa by pushing on the transfer pipet. The weck-cel sponges were left to absorb mucosal fluid for 5 min before pulling them back into the pipet and removing the pipet from the rectum. Weck-cel sponges were then placed into 5 mL polypropylene tubes on dry ice.

#### 2.4.4. Detection of IFN-γ Producing Cells by ELISpot Assay

The day before the assay, PVDF-membrane plates (Merck Millipore, Burlington, MA, USA) were coated with 50 µL of 5 µg/mL monoclonal antibody anti-IFN-γ (clone 1-D1K or GZ-4, Mabtech, Mariemont, OH, USA), and incubated overnight at 4 °C. Plates were washed 4 times with complete RPMI and then incubated with fresh complete RPMI for a minimum of 30 min at 37 °C. Two hundred thousand cells were added to the wells and stimulated with 2 µg/mL of peptide pools spanning SIV-Gag, HIV-Env C-terminal or SIV-Nef (AIDS Reagent Repository, NIH-NIAID catalog numbers 6204, 6451, and 8762, respectively). Medium only or PMA-Ionomycin were used as negative and positive controls, respectively, and incubated for 20 h at 37 °C. Plates were washed 4 times with PBS + 0.05% Tween-20 and captured IFN-γ spot-forming units were revealed using anti-IFN-γ secondary antibody coupled with biotin, then incubated with Avidin-D-HRP. Spots were revealed using TMB (SeraCare, Milford, MA, USA), washed with tap water and counted with a CTL-6 Immunospot reader (Cellular Technology Limited CTL, Shaker Heights, OH, USA) after drying the plates.

#### 2.4.5. Multiparametric Flow Cytometry Assay

Mouse splenocytes, macaque PBMCs and lymph node cells were isolated as previously described [24,34]. To assess the phenotype and cytokine production of vaccine-specific T cells, one million cells were stimulated for 6 h with 2 µg/mL of each peptide pool (SIV-Gag, HIV-Env or SIV-Nef, (AIDS Reagent Repository, NIH-NIAID catalog numbers 6204, 6451, and 8762, respectively)) or with PMA/Ionomycin as positive control. Non-stimulated cells were used to subtract the basal secretion signal. For mice splenocytes, 35 μ g/mL of TAPI-2 (Sigma, Burlington, MA, USA) was added after 1 h incubation with peptide pools to avoid the CD62L cleavage. CD28 and CD49d (0.5 µg/mL) were used for co-stimulation with Brefeldin A to prevent cytokine excretion. Cells were washed and stained for 20 min with viability marker (Live/Dead, Invitrogen, Waltham, MA, USA) and 30 min with surface Abs for surface staining. CD16/CD32 was added in mouse splenocyte suspensions for 15 min at 4 °C after the live/dead staining to block the Fc receptor. Cells were then permeabilized with BD Cytofix/Cytoperm reagent (BD Biosciences, San Jose, CA, USA) for 20 min at 4 °C and stained for intracellular compounds for 30 min. Antibodies were purchased from BD and eBioscience (Table 1 and Table 2). Analysis was performed on a four-laser BD Aria Fusion instrument, calibrated daily with CS&T beads (BD), compensated with anti-Ig κ (BD CompBeads positive and negative) and ArC Beads (Invitrogen, Waltham, MA, USA). Results were analyzed with FlowJo software for Windows (Version 10.7.1).

#### 2.4.6. Detection of Total and HIV-Specific Antibodies

ELISA plates (Nunc MaxiSorp™ flat bottom, waltham forest, MA, USA) were coated with anti-Ig or antigen to test plasma or rectal secretion in order to detect total Ig or specific IgG and IgA. For total Ig, plates were coated with 0.25 µg of polyclonal goat anti-monkey IgG + IgA + IgM (LSBio, Seattle, WA, USA) in 100 µL PBS, for 1 h at room temperature. For specific Ig, plates were coated with 2.5 µg Concanavalin A (ConA) (Sigma, Burlington, MA, USA) diluted in 100 µL of 10 mM HEPES buffer, then incubated for 1 h with HIV-1_JR-FL_ virus diluted in PBS + 1% triton X100 (Sigma, Burlington, MA, USA). Plates were blocked in 2% non-fat dry milk in PBS (Blotting-grade Blocker, BioRad, Hercules, CA, USA) at 4 °C overnight. Dilution of 1:4 (mice) and serial dilutions from 1/50 to 1/6250 (macaques) in PBS 2% non-fat dry milk were used to determine the OD or titers of Ig in the serum samples for each time point. Secondary antibody (Goat Anti-Monkey or Goat Anti-Mouse IgG gamma Horseradish Peroxidase (HRP) Conjugated, (Rockland, Limerick, PA, USA) or goat Anti-Monkey IgA Biotin Conjugated, (Rockland, Limerick, PA, USA) plus Avidin-D HRP (Vector Laboratories, Burlingame, CA, USA) were used. TMB (SeraCare, Milford, MA, USA) was added to the plate, and the reaction was stopped with 1M sulfuric acid and immediately read at 450 nm with an ELISA reader (Synergy HT, Bio-Tek, Taunton, MA USA).

#### 2.4.7. Neutralization Assay

The TZM-bl cell line [35] was used for the neutralization assay. Briefly, heat-inactivated macaque plasma samples taken before vaccination (week 0) and after boost (week 18) were prepared at five-fold serial dilutions starting at 1:10. The diluted plasma samples were incubated with HIV-1_JR-FL_ Tier 2 virus (Clade B) for 1 h at 37 °C and then used to inoculate TZM-bl cells. The luciferase activity was measured 48 h post infection. The definition of 50% inhibitory dose (ID50) reported as plasma reciprocal dilution was the sample dilution at which relative luminescence units (RLU) were reduced by 50% compared with RLU in virus control wells after subtraction of background RLU in cell control wells.

#### 2.4.8. Antibody-Dependent Cellular Cytotoxicity (ADCC) Assay

The ADCC assay was based on a previously described method [36]. The target cells used were CEM.NKr-CCR5 CD4^+^ T cells expressing firefly luciferase upon HIV-1 infection. NK cells KHYG-1 expressing CD16 served as effector cells. To measure ADCC, target cells were infected with 50 ng of HIV-1_JR-FL_ virus by spinoculation and cultured for 4 days. Two-fold serial dilutions of each heat-inactivated plasma sample were added to the infected targets for 20 min at room temperature in round-bottom 96-well plates. KHYG-1 effector cells were added at a 10:1 effector-to-target ratio and incubated for an additional 8 h. The cells were lysed, and RLU was measured immediately on a BioTek luminometer. ADCC activity was calculated as follows: (mean RLU at given plasma dilution − mean background RLU)/(mean maximal RLU − mean background RLU) × 100. Fifty percent (50%) ADCC titers indicate the plasma titers required for half-maximal cell lysis (similar to 50% inhibitory concentration (IC_50_) for neutralization).

#### 2.4.9. Statistical Analysis

Statistical analysis was determined using GraphPad Prism 8 software (version 8.4.3). Medians and means were used for the different graphic presentations. Non-parametric Mann–Whitney, parametric one-way ANOVA and Tukey’s multiple comparison tests were used to determine the significance.

## 3. Results

### 3.1. CAL-SHIV-IN^−^ IRES-Cytokine Prototypes Produce Functional Viral Proteins

CAL-SHIV-IN^−^ IRES-cytokine vaccine vectors (Figure A1) were transfected into HEK 293T cells using the phosphate calcium method to test whether the vaccine proteins were expressed. Transfected plasmids expressing the macaque cytokines alone and non-transfected (NT) HEK 293T cells were used as negative controls (Figure 1a). Twenty-four hours post-transfection, supernatants were removed, and cell monolayers were washed twice to remove any residual plasmid. Non-adherent indicator, highly fusogenic M8166 CD4 T cells were then seeded on the monolayers to detect cytopathic effects (CPEs) as giant multinucleated cells following fusion of infected M8166 cells. Control co-cultures of non-transfected HEK 293T or HEK 293T cells transfected with pNDgms-rmIL-15 showed no CPE (Figure 1a,b). In contrast, CPEs were readily visible in co-cultures of HEK 293T cells transfected with CAL-SHIV-IN^−^ IRES-cytokine constructs (Figure 1c). This demonstrated that the viral Env glycoproteins were correctly synthetized, matured and released or exposed on the surface of transfected cells.

Supernatants from control and CAL-SHIV-IN^−^ IRES-cytokine transfected HEK 293T cells were also harvested 3 days post-transfection and used to inoculate the permissive M8166 cell cultures. Again, CPEs were evidenced by 24 h post-infection only with supernatant of HEK 293T cells transfected with CAL-SHIV IN^−^ IRES-cytokines (Figure 1f, black arrows). While the CPEs shown in co-cultures resulted mainly from cell-to-cell contacts (Figure 1c), here the CPEs resulted from direct infection with cell-free one-cycle viral particles released and harvested in the supernatant of transfected cells.

SIV Gag p27 proteins expressed and released from transfected HEK 293Ts were also quantified during three successive days. The highest Gag p27 production was seen on day 3 post-transfection for each of the 3 vectors, with 991 pg/mL for CAL-SHIV-IN^−^, 1476 pg/mL for CAL-SHIV-IN^−^ IRES IL-7 and 823 pg/mL for CAL-SHIV-IN^−^ IRES IL-15 (Figure 1g). The Gag p27 expression levels were comparable for the 3 plasmids, suggesting that the insertion of the IRES-cytokine cassette into the genome of CAL-SHIV-IN^−^ did not impact Gag expression.

### 3.2. IL-7 and IL-15 Cytokines Inserted into CAL-SHIV-IN^−^ Genome Are Expressed, Secreted and Functional

Expression and secretion of both cytokines inserted into the parental vector (Figure A1) were assessed by ELISA in the supernatants of transfected HEK 293T cells. Unlike Gag p27, the maximum cytokine expression and release occurred on day 2 post-transfection and decreased thereafter (Figure 2). The concentration of IL-7 released on day 2 post-transfection was similar in cultures transfected with CAL-SHIV-IN^−^ IRES IL-7 alone (1664 pg/mL) and those co-transfected with CAL-SHIV-IN^−^ IRES IL-15 (1583 pg/mL) (Figure 2A), indicating that co-transfection had no impact on IL-7 synthesis and release. Similar results were obtained for the production of IL-15 (Figure 2B), suggesting an absence of interference between both transfected constructs. As expected, the control cassettes of IRES-cytokines in antisense did not produce any detectable cytokine.

The cytokine functions were also tested for their ability to drive the homeostatic differentiation of precursors using a functional assay developed previously in our laboratory [23]. PBMCs from a BCG immunized donor were used as target for cytokine stimulation. Immediate effector cells and PHPC-derived effector memory cells were quantified in duplicates of three independent experiments by IFN-γ ELISpot at day 1 and day 12, respectively. The PHPC assay showed an increase in effector memory T cells in all tested conditions (Figure 2C). The recombinant cytokines, used as positive control, induced the highest increase with a median of 1700 spots/million PBMC after 12 days of restimulation with the BCG-vaccine antigen, while only a median of 59 spots/million PBMCs was observed on day 1. Cells cultured with the supernatants of transfected cells demonstrated an increase in numbers, reaching 1000 spots/million PBMCs after 12 days of culture with supernatants from the pNDgms-rmIL-7 and pNDgms-rmIL-15 plasmid transfections, and 750 spots/million PBMCs with supernatants from the CAL-SHIV-IN^−^ IRES IL-7 and CAL-SHIV-IN^−^ IRES IL-15 DNA vaccines. The differences observed between recombinant cytokines and those contained in supernatants were likely associated with the purity of the commercial cytokines. Nonetheless, the expressed cytokines allowed for differentiation and proliferation of PBMCs, clearly demonstrating the functionality of the cytokines released by both CAL-SHIV-IN^−^ IRES-cytokine vectors.

### 3.3. Vaccine-Specific T Cell Responses in Vaccinated Mice

T cell immune responses evaluated in splenocytes of immunized mice are summarized in Figure A3. Responses were undetectable following the first immunization (Figure 3). However, Gag- and Nef-specific responses started to be detectable 2 weeks after the boost (week 8 post prime-immunization). Interestingly, 4 weeks after the boost, a marked difference was observed between the two groups of mice. Indeed, mice immunized with CAL-SHIV-IN^−^ showed low CD4 responses to Gag and Nef. CD8 responses were generally undetectable in this group. In contrast, mice immunized with the cytokine-expressing vectors displayed much higher proportions of IFN-γ and IL-2 producing CD4 and CD8 cells specific to both Gag and Nef.

### 3.4. Env-Specific Antibody Responses in Vaccinated Mice

HIV Env-specific IgG measured in serum samples is presented in Figure 3C. At two weeks post-immunization, no IgG was detected for both groups 1 and 2. In contrast, a significant (*p =* 0.0160) increase in IgG production was detected in group 1 at week 8 (2 weeks post-boost). A significant (*p =* 0.0382) decrease in IgG titers was observed in group 1 at week 10. The profile of antibody responses did not show significant changes between the 3 time points (weeks 2, 8 and 10).

### 3.5. Evaluation of Vaccine-Specific T Cell Responses in Immunized Rhesus macaques

The increased immunogenicity in mice guided the immunization protocol for the macaques. Rhesus macaques received two immunizations at a 16-week interval (Figure 4A), with combined intramuscular and intradermal injections, both with EP, as described in the materials and methods section. The vaccine-induced T cell responses were analyzed by IFN-γ ELISpot and multiparametric flow cytometry up to 40 weeks post-immunization.

#### 3.5.1. IFN-γ ELISpot Responses

Macaque PBMCs were tested longitudinally for T cell responses by ELISpot (Figure 4A). Gag (median 77.5 (group 1) and 37.5 (group 2) spots/million PBMCs) and Nef (median 10 (group 1) and 12.5 (group 2) spots/million PBMCs) were detectable as early as 2 weeks post-immunization, but not Env-specific responses. Intriguingly, Gag but not Nef responses appeared boosted in both groups. Interestingly, a peak of Gag-specific IFN-γ secreting cells was observed at week 26 for group 2, reaching 167.5 spots/million PBMCs (near 5-fold increase), suggesting that the IL-7 and IL-15 adjuvanted group 2 may have had a higher recall of effector Ag-specific T cells compared to group 1 (Figure 4A).

#### 3.5.2. Intracellular Cytokine Analysis

To further characterize the Ag-specific T cells post-immunization, we performed phenotype/function analysis of PBMCs at selected time points (Figure 4B) and of mononuclear cells from draining lymph nodes (Figure 4C). Cells were phenotyped and tested for the production of IFN-γ, TNF-α, MIP-1β, and Granzyme B as well as the degranulation factor CD107a (Figure A4). Gag-stimulated PBMCs at weeks 26 and 36 post-immunization (10- and 20-weeks post-boost, respectively) were normalized both with unstimulated cells (medium alone) and by subtracting the background responses of each individual macaque before immunization (week 0). Data in Figure 4B show an increase in both CD4 and CD8 T cells producing the inflammatory cytokines after the boost. Surprisingly, the proportions of these cytokine-producing cells were higher for macaques immunized with the parental plasmid (non-adjuvanted with cytokine). Granzyme B secretion was higher at 36 weeks, whereas MIP-1β was expressed at higher levels around week 26, for both CD4 and CD8 cells.

In the draining lymph nodes, CD4 and CD8 T cells from weeks 18 to 36 showed maintenance of Gag-specific memory T cells, and their capacity of recall. Correlating with PBMC responses, the proportions of cytokine-secreting cells were higher for group 1 macaques. However, at week 18, the recall of specific CD8 T cells from group 2 macaques led to higher proportions of cells secreting Granzyme B and those expressing CD107a (Figure 4C).

### 3.6. Evaluation of Vaccine-Specific Antibody Responses in Rhesus Macaques

#### 3.6.1. Evaluation of Env-Specific Abs in Macaque Serum Samples

Serum samples of immunized macaques were titrated for Env-specific IgG against HIV-1_JR-FL_ virus antigens by limiting dilution ELISA (Figure 5). Env-specific antibodies were detectable in both groups after the first immunization, and the titers increased after the boost. These titers were maintained in both groups until week 22 (6 weeks post-boost). However, by week 40 (>5 months after the boost), antibody titers diverged between the 2 groups. While high titers were maintained in group 2, they markedly decreased in group 1, suggesting a role for IL-7 and IL-15 in the homeostasis/maintenance of plasma cells, enhancing memory humoral responses. However, these antibodies failed to translate into any neutralizing responses against HIV-1_JR-FL_ tier 2 virus in the serum samples of any macaque (Figure 5B).

Aside from neutralizing activity, ADCC represents another mechanism able to restrict HIV replication, with ADCC titers correlated with viral control in both macaques and humans [37]. Serum samples of vaccinated macaques were tested for ADCC activity with infectious HIV-1_JR-FL_ virus. Interestingly, Env-specific IgG of both groups showed ADCC activities at weeks 22 and 40 (Figure 5C). However, there was no statistical difference between the values obtained for the two groups. In contrast, a positive correlation was determined between ADCC activity and specific IgG binding titer (Figure 5D). This correlation was stronger for groups 1 and 2 at week 22 (*r* = 0.86 and 0.99, respectively) than at week 40 (*r* = 0.77 and 0.70).

#### 3.6.2. Evaluation of Env-Specific IgG and IgA in Rectal Secretions

Since HIV-1 is mainly transmitted by sexual intercourse (via infected semen, vaginal secretions, or infected cells), mucosal antibodies are thought to be the first line of defense to counteract viral infection. Rectal secretions from vaccinated macaques were collected at different time points and analyzed for the presence of mucosal Env-specific antibodies. Rectal secretions were assayed by ELISA against HIV-1_JR-FL_ virus to detect Env-specific IgG and IgA (Figure 5E). Mucosal Env-specific IgG responses were only detected post-boost in both groups (weeks 18 and 22). Interestingly these IgG responses, while contracting from week 26 to 32, reappeared at weeks 36 and 40 but only in group 2. In contrast, mucosal IgA responses were only detectable in rectal secretions of group 2 but not group 1 macaques. These IgA responses were first detected after the prime immunization, followed by a rapid decline. However, the second immunization yielded a prolonged anamnestic response 16 weeks later. Moreover, IgA responses were again detectable after a contraction phase of a minimum of 6 weeks (Figure 5E). This suggests that the expression of both cytokines by the vaccine enhanced mucosal immune responses.

## 4. Discussion

It is well-established now that IL-7 and IL-15 cytokines regulate memory T cell homeostasis. These members of the gamma-chain (γ-c) cytokine family maintain, expand, differentiate and activate T cells to fight chronic infections, (reviewed in [25]). During HIV infection, these cytokines positively regulate both HIV-specific CD4 and CD8 T cells [38,39]. While this positive regulation of CD8 is beneficial for the control of HIV-1 replication, activation of memory CD4 T cells has the potential to increase the number of activated HIV-1 targets [38,40]. In our previous study, we reported that macaques immunized with an HIV lentiDNA vaccine had mounted vaccine-specific T cells that lasted for at least 80 weeks after a single immunization. Detailed characterization of these cells revealed that CD8 T cells were composed of both central and effector memory T cells, while CD4 T cells comprised mainly central memory T cells [23]. Interestingly, these T cell responses correlated with persistent control of mucosally administered SIV_mac251_ pathogenic virus. This control was obtained in the absence of detectable neutralizing antibodies [24]. In the present study, we evaluated the immunogenicity of this SHIV-based lentiDNA vaccine, which was adjuvanted with two homeostatic cytokines: IL-7 and IL-15. Our aim was to create favorable conditions for antigens and cytokines to elicit a high proportion of T cells with augmented potential for homeostasis and the generation of highly functional effector memory T cells. We found that our lentiDNA vaccine prototypes did efficiently co-express both vaccine antigens HIV-Env, SIV-Gag p27 (1476 for CAL-SHIV-IN^−^ IRES IL-7 and 823 pg/mL for CAL-SHIV-IN^−^ IRES IL-15) and the cytokines IL-7 and IL-15 (1654 and 125 pg/mL, respectively), which were released in the supernatant of transfected HEK 293T cells (Figure 1c,f and Figure 2A,B). These lower concentrations of cytokines are compatible with physiologic functions but not with cytokine storm-like induction [41]. To evaluate the function of IL-7 and IL-15 released in the supernatant of transfected cells, we used our lab-adapted PHPC assay on human PBMCs. We compared the activity of recombinant cytokines with that of cytokines harvested from the supernatants of transfected HEK 293T cells. Even though cytokines from the supernatants were not purified or concentrated, they increased the proportion of T cells by at least ten-fold as well as their long-term survival, confirming their functionality *in cellulo*. IL-7 therapy in SIV-infected macaques increased CD4 and CD8 T-naïve and memory subsets not only in peripheral blood but also in other compartments such as spleen, lymph nodes, lungs and kidneys, and facilitated their maintenance [42]. Similarly, IL-15 therapy in SIV-infected macaques under ART showed a proliferation of CD4 and CD8 Tem, able to disperse in extra-lymphoid compartments [43]. Additionally, ART-treated SIV_mac251_-infected macaques multiply immunized with a DNA vaccine expressing SIV antigens along with IL-15 and IL-15Rα elicited Ag-specific responses with both Tcm and Tem associated with inflammatory cytokine secretions. Interestingly, after ART interruption, vaccinated monkeys showed long-lasting partial control of the viremia, with one log decrease, compared to unvaccinated macaques who developed AIDS after ART interruption [44]. These studies support the hypothesis that IL-7 and IL-15 cytokines, associated with our lentiDNA vaccine, have the potential to enhance the pool of Ag-specific memory CD4 and CD8 T cells.

DermaVir vaccine comprised a pDNA expressing 15 HIV-1 antigens. This vaccine has been adjuvanted with IL-7 or IL-15 and used as a therapeutic vaccine. This vaccine was used to expand the HIV-specific memory T cell pools in patients already exposed to HIV-1 and developed both antibody and T cell responses. DermaVir-vaccinated patients failed to induce responses potent enough to fully suppress viral infection. DermaVir prototype was similar to our vaccine, except the genome was controlled by Tat-dependent SIV/HIV LTRs and ours, by the constitutive CAEV LTRs. Our vaccine prototypes expressed concomitantly the cytokine adjuvants with the vaccines antigens, while DermaVir did not. Gag pDNA delivered as DermaVir in mice significantly enhanced Gag-specific memory responses [29]. Similarly, SHIV-based pDNA delivered as DermaVir in rhesus macaques significantly enhanced Gag- and Env-specific central memory and effector memory T cells, with Th1 and Th2 cytokines, similarly to our lentiDNA vaccine [45].

BALB/cJ mice immunized with cytokine-adjuvanted lentiDNA developed higher frequencies of inflammatory cells secreting both IFN-γ and IL-2 starting 2 weeks after the boost (week 8 post-immunization), compared to mice immunized with the parental vaccine. This result suggested that co-expression of both IL-7 and IL-15 cytokines enhanced the elicitation of multifunctional antigen-specific T cells. Another study demonstrated that a DNA-based vaccine expressing the SIV-gag gene led to Ag-specific CD4 and CD8 T cells in vaccinated IL-15 KO mice, but CD8 absolute numbers were lower compared to those in wild-type mice [46]. In addition, these CD8 T cells had impaired cytotoxicity, as Granzyme B secretion was reduced compared to that in wild-type mice. This study highlighted that IL-15, in addition to DNA-based vaccine, is largely involved in the generation and maintenance of CD8 T cell responses as well as critical for their number, activity and function [46]. Finally, another recent intriguing report has highlighted the long-term protection afforded by deglycosylated SIV-Env immunization, which was associated with IL-15-mediated effector functions of T and NK cells [47].

Surprisingly, mice immunized with the parental lentiDNA, but not the cytokine-adjuvanted lentiDNA, produced HIV Env-specific IgG. This could have resulted from a shift of balance from Th2 to Th1 responses. In addition, since cytokine adjuvants were originated from rhesus macaques, their functions might not be compatible with mouse B cells.

The prime immunization of macaques with the parental and the cytokine-expressing lentiDNA vaccines did not show a detectable difference, as the responses were low. However, as in mice, the boost led to increased Gag-specific IFN-γ producing T cells in macaques immunized with the adjuvanted but not the parental lentiDNA vaccine. Because Gag-conserved epitopes have been described among individuals infected with HIV-1, this conserved protein is a promising target for an HIV vaccine [48]. In contrast, no detectable Env-specific IFN-γ producing T cells were found in macaques immunized with the parental or adjuvanted lentiDNA vaccine.

Interestingly, longitudinal characterization of antigen-specific CD4 and CD8 T cells showed a different evolution. While increased multifunctional CD8 T cells were detected in macaques immunized with the adjuvanted lentiDNA vaccine at late time points, no detectable increase was seen in the CD4 population. These findings confirmed those from our earlier studies showing that immunization of macaques with a single dose of lentiDNA elicited long-term central and effector memory CD8 T cells but only central memory CD4 T cells [23].

The production of antiviral cytokines such as IFN-γ, Granzyme B and MIP-1β in macaques after the boost is particularly important for viral control and could lead to protection during a primary infection by killing infected cells before the setup of a viral reservoir. Moreover, a study has demonstrated that the induction of a combination of degranulation factor CD107a and MIP-1β responses led to the inhibition of HIV-1 infection [49]. This phenomenon was also found in HIV-1 elite controllers [49].

The longitudinal evaluation of vaccine-specific antibodies against Env in both groups showed similar titers of peripheral IgG up to week 22 post-immunization. However, at week 40, a marked decrease in the Env-IgG titer was observed in the parental, but not in the adjuvanted, group of vaccinated macaques, suggesting that the cytokines helped to maintain plasma cells secreting Env-specific IgG during the memory phase, weeks 22–40 post-immunization. This humoral memory is important for the long-term control of chronic infectious disease. While this IgG response was not associated with detectable neutralizing activity against Tier-2 HIV-1_JR-FL_, it exhibited clear ADCC activity, a function associated with HIV control [44]. As demonstrated in a number of macaque studies, neutralizing Abs are not essential to protect against SIV and, by extension, HIV-infection [23,24,50,51]. Vaccine-induced non-neutralizing serum and mucosal Abs with ADCC activity have shown inhibition of the infection after intrarectal challenge with SIV_mac251_ [23,24,50] or with SHIV89.6P [51]. This protection was also validated in humans, as ADCC activity was also found in HIV-1 infected non-progressor patients and one of the correlates of protection in the RV144 clinical trial [52]. Moreover, breast milk non-neutralizing Abs with ADCC capacity have been correlated with reduction of vertical transmission from viremic mothers to infants [53].

HIV is transmitted mucosally. IgA is important for the control of this type of transmission, and recent findings demonstrated that both IgA and IgG can block HIV-1 infection after mucosal exposure [54]. We evaluated the proportion of rectal IgA and IgG in vaccinated macaques. While in macaques immunized with the parental, non-adjuvanted prototype, IgA titers were essentially undetectable in rectal secretions, macaques immunized with the adjuvanted prototype showed clearly detectable HIV Env-specific IgA titers (up to 1:250) at multiple time points. By comparison, antigen-specific mucosal IgG titers were only transiently detected after the boost in the adjuvanted lentiDNA group but not the parental prototype. In addition, IgA responses were found augmented at later time points following a contraction period, a phenomenon reported by others [55]. We did not expect to find these humoral response discrepancies between mice and macaques. However, in our earlier studies, T cell but not antibody responses were readily detected in BALB/c mice immunized with lentiDNA prototype [34,56]. Cytokine co-expression with lentiDNA vaccine Ag did not boost the humoral responses.

In summary, our CAL-SHIV-IN^−^IRES-cytokine lentiDNA vaccine and vaccination strategy elicited potent vaccine-specific, non-neutralizing antibodies with ADCC capacity in the periphery. Furthermore, a strong IgA response was observed in mucosal compartments of rhesus macaques. These humoral responses were durable, up to 40 weeks post-immunization (the end of monitoring) and were enhanced by the cytokines. In addition, while cellular responses to Env and Nef were minimal, Gag-specific central memory and effector T-cell responses secreting Granzyme B and MIP-1β were clearly elicited. Future studies will be needed to test whether such responses can contain challenges with pathogenic SHIVs to provide definite proof of the protective capacity of the immunity conferred by these vaccines and this original strategy.

## Figures and Tables

**Figure 1 vaccines-10-00461-f001:**
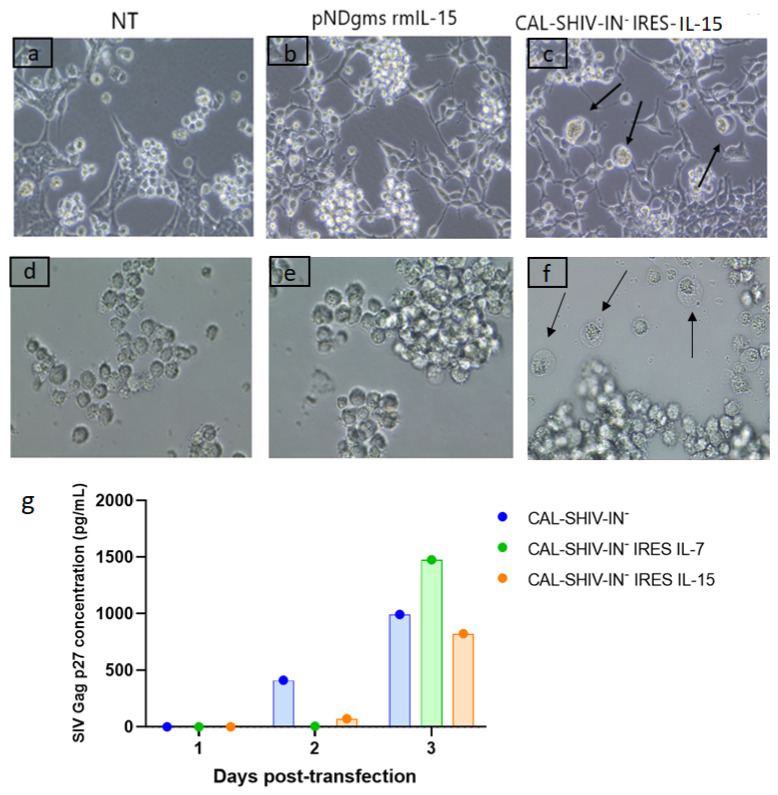
Functional expression of SIV proteins. Co-culture of permissive M8166 lymphocyte cells with HEK 293T cells non-transfected (**a**), transfected with pNDgms-rmIL-15 (**b**) or transfected with CAL-SHIV-IN^−^ IRES-cytokines (**c**). CPEs were observed in the cultures with the CAL-SHIV-IN^−^ IRES-cytokines (black arrows). Supernatants of HEK 293T cells in (**a**–**c**) were used to inoculate cultured M8166 in (**d**–**f**), respectively. At 24 h post-inoculation, CPEs were observed only in (**f**) (black arrows). (**g**) Quantification of SIV Gag p27 by ELISA: Supernatants of HEK 293T cells in a, b and c harvested at days 1, 2 and 3 post-transfection were analyzed by ELISA. Results were expressed in pg/mL.

**Figure 2 vaccines-10-00461-f002:**
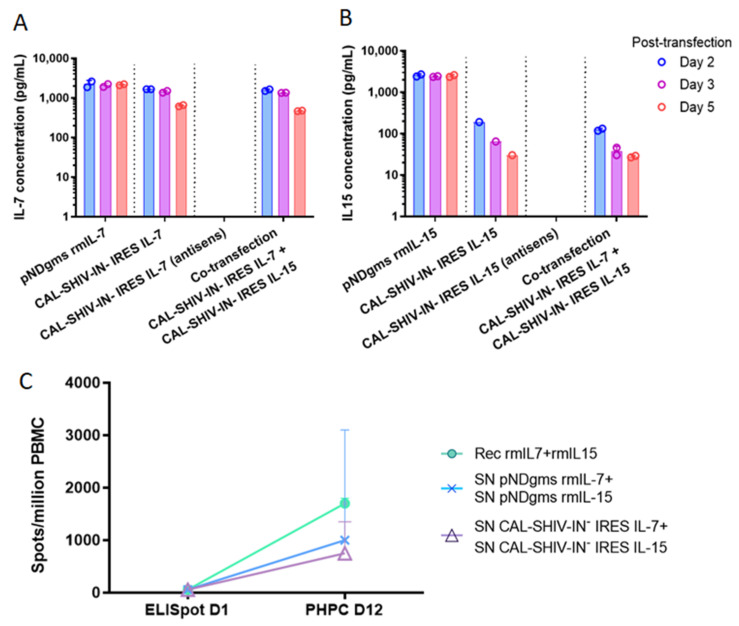
Expression and functional analysis of secreted IL-7 and IL-15 cytokines: HEK 293T cells were transfected with either pNDgms-rmIL-7, pNDgms-rmIL-15 or with CAL-SHIV-IN^−^ IRES-rmIL-7 and CAL-SHIV-IN^−^ IRES-rmIL-15 alone or co-transfected with both CAL-SHIV-IN^−^ IRES cytokines. Vaccines with antisense cytokine cassette were used as negative controls. Supernatants of transfected cells were harvested at days 2, 3 and 5 and quantified for IL-7 (**A**) and IL-15 (**B**). Median values of two independent experiments were used to draw the columns. (**C**) Human PBMCs from a BCG-vaccinated donor were used in overnight and PHPC assay using the homeostatic cytokines. Quantified spots after 20 h stimulation with BCG vaccine are noted as ELISpot day 1. Spots quantified at 12 days stimulation with indicated cytokines (**–**o**–**): recombinant cytokines; (**–**x**–**): supernatants of pNDgms transfected HEK 293T cells; (**–**Δ**–**): supernatants of CAL-SHIV-IN^−^ IRES cytokines transfected HEK 293T cells. Median values with 95% CI of the 3 repeated experiments (day 12) are represented.

**Figure 3 vaccines-10-00461-f003:**
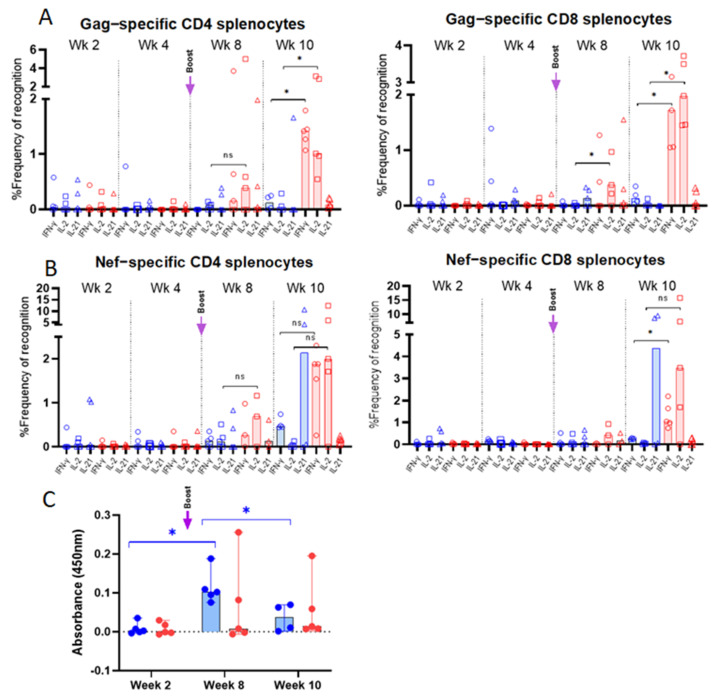
Flow cytometry characterization of vaccine-specific T cells in spleens of BALB/cJ-immunized mice: Splenocytes from group 1 (CAL-SHIV-IN^−^; in blue) and group 2 mice (CAL-SHIV-IN^−^ IRES-rmIL-7 + CAL-SHIV-IN^−^ IRES-rmIL-15; in red), harvested at different time points post-immunization, were stimulated with SIV-Gag and -Nef peptides. Cells were stained with the combination of Abs reported in Table 1, to identify Gag (**A**) and Nef (**B**) specific CD4 and CD8 cytokine-secreting T-cells. Results were normalized with unstimulated splenocytes to remove the basal secretion, and median responses were plotted. Non-parametric Mann–Whitney test was calculated. Circles represent the percentage of IFN-γ secreting cells, squares the percentage of IL-2 secreting cells, and triangles the percentage of IL-21 secreting cells. (**C**) HIV Env-specific IgG in mouse serum: Serum samples from the two groups were analyzed by ELISA at 2, 8 and 10 weeks post-immunization for the detection of Env-specific IgG. O.D. (450 nm) are presented after blank and baseline subtraction. Median values with 95% CI are shown. Parametric Tukey’s multiple comparison test was calculated. Significant increase (*p* = 0.0160) was observed between weeks 2 and 8, and decrease (*p* = 0.0382) was observed between weeks 8 and 10 for group 1. No statistical significance was observed for group 2. ns: not significant; *: *p* < 0.05.

**Figure 4 vaccines-10-00461-f004:**
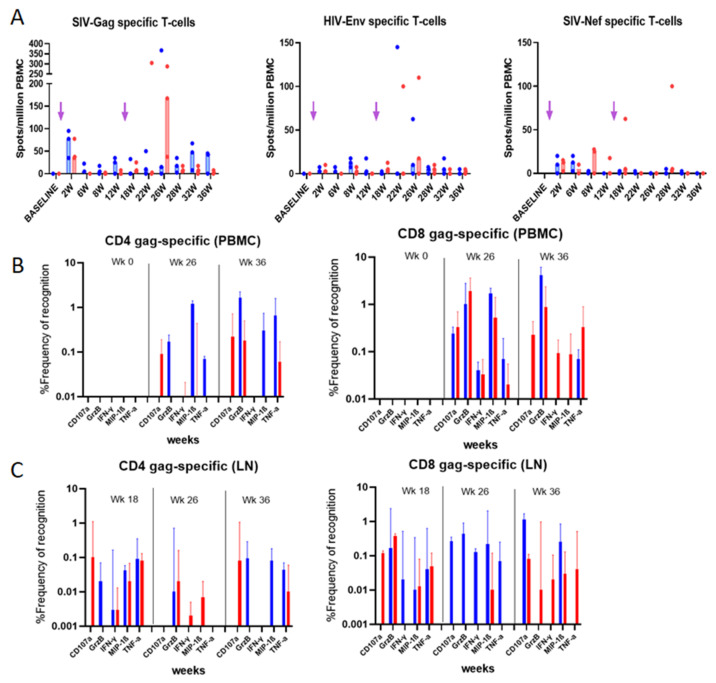
Vaccine-specific T-cell responses in blood and LN mononuclear cells from immunized rhesus macaques: (**A**) IFN-γ ELISpot responses to Gag, Env and Nef following overnight stimulation. Unstimulated cells were used as negative control for normalization. Median of the 3 animals per group is represented. Arrows indicate immunization times. (**B**) PBMCs and (**C**) LN mononuclear cells were used for the ICS assay using the panel of Ab reported in Table 2. Frequencies of Gag-specific cells measured at indicated time points are reported. Median values with 95% CI of the proportion of cytokine-secreting cells are shown. CAL-SHIV-IN^−^ group is presented in blue, and CAL-SHIV-IN^−^ IRES-rmIL-7 + CAL-SHIV-IN^−^ IRES-rmIL-15 in red.

**Figure 5 vaccines-10-00461-f005:**
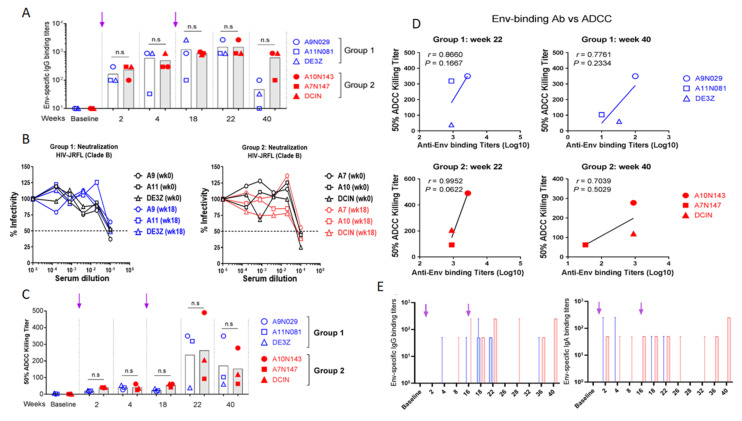
Vaccine-specific Abs in sera and mucosal secretions of vaccinated rhesus macaques. (**A**) Serum IgG samples harvested at indicated time points were evaluated by ELISA against inactivated HIV-1_JR-FL_ virus. Mean titers are plotted following subtraction of blank and baseline values. Arrows indicate immunization times. (**B**) Neutralizing activity of serum Abs was evaluated using the serum neutralization assay against Tier 2 HIV-1_JR-FL_ virus and TZM-bl indicator cells. IC_50_ was determined by measuring the luminescence after 48 h incubation. (**C**) Evaluation of ADCC activity. Sera from both groups of vaccinated monkeys were incubated with HIV-1 _JR-FL_ to inoculate CEM.NKr-CCR5 CD4^+^ target T cells. Effector KHYG-1 NK cells were added at a 10:1 effector-to-target ratio. After cell lysis, RLU was measured and ADCC activity evaluated for IC_50_. Mean values are reported. (**D**) Comparative evaluation of serum-binding and ADCC specific to Env epitopes in vaccinated macaques. Values were plotted to determine the correlate. (**E**) Evaluation of Env-specific Abs in rectal secretions. Rectal secretions were collected at indicated time points and analyzed by ELISA to detect both HIV-specific IgG and IgA against inactivated HIV-1 _JR-FL_. Cutoff was determined by subtracting the OD values of blank and baseline and median titers with 95% CI are shown. Group 1 (CAL-SHIV-IN^−^) is represented in blue, and group 2 (CAL-SHIV-IN^−^ IRES-rmIL-7 + CAL-SHIV-IN^−^ IRES-rmIL-15) in red. ns: not significant. Arrows indicate immunization times.

**Table 1 vaccines-10-00461-t001:** Panel of anti-mouse antibodies used for the ICS assay.

Extracellular Markers	Fluorochrome	Clone	Company
CD3	Alexa Fluor 700	17a2	BD Pharmingen
CD4	APC-H7	GK1.5	BD Pharmingen
CD8a	Brilliant Violet 605	53–6.7	BD Horizon
CD127	PE-Cy7	Sb/199	BD Pharmingen
CD62L	PE-CF594	Mel-14	BD Horizon
**Intracellular Markers**	**Fluorochrome**	**Clone**	**Company**
IFN-γ	Brilliant Violet 711	Xmg1.2	BD Horizon
IL-21	eFluor450	Ffa21	eBioscience
IL-2	APC	Jes6-5H4	eBioscience

**Table 2 vaccines-10-00461-t002:** Panel of anti-monkey antibodies used for the ICS assay.

Extracellular Markers	Fluorochrome	Clone	Company
CD3	Alexa Fluor 700	SP34-2	BD Pharmingen
CD4	FITC	L200	BD Pharmingen
CD8	APC-Cy7	SK1	BD Pharmingen
CD28	ECD	n/a	IO Test
CD95	PerCp-Cy5.5	DX2	BD Pharmingen
CD45	APC	D058-1283	BD Horizon
CD107a	BV650	H4A3	Biolegend
**Intracellular Markers**	**Fluorochrome**	**Clone**	**Company**
IFN-γ	PE-Cy7	B27	BD Pharmingen
TNF-α	BV785	Mab11	Biolegend
MIP-1β	V450	D21-1351	BD Horizon
Granzyme B	PE	GB11	BD Pharmingen
Brilliant buffer plus			BD horizon

## Data Availability

Data are archived in PAVAL Lab. and the NIRC servers. https://www.pavallab.com/ and https://nirc.louisiana.edu/.

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
