# Peer review of "Cytokine Adjuvants IL-7 and IL-15 Improve Humoral Responses of a SHIV LentiDNA Vaccine in Animal Models"

_vaccines, 2022, doi:10.3390/vaccines10030461_

Round 1

Reviewer 1 Report

In this manuscript, the authors have extended their previous work on developing a SHIV based lentiDNA vaccine against HIV. Previously, they have established a single dose, integration deficient vaccine containing SHIV genome except for the region responsible for integration.

In this work, they have enhanced their vaccine to include a IL7 and IL15 to improve T cell mediated responses . They show that both cytokines were secreted from transfected cells at physiologic concentrations, resulting in increase in proportion and survival of T cells without  impeding expression of viral antigens. In vivo characterization of this vaccine included both mouse and a non-human primate model. In mice, the vaccine adjuvanted with IL-7 and IL-15 resulted in increased secretion of IFN-g and IL-2. Similarly, increased Gag-specific T cells were observed in Macaques vaccinated with the cytokine containing vaccine. The adjuvanted vaccine also caused increased secretion of several antiviral cytokines, and long-term humoral responses.

This manuscript is well written, experiments well thought out and with appropriate controls.

Only minor adjustments should be considered before publication.

  1. What is the advantage/difference of this vaccine over the DermaVir, a HIV-1 Gag based vaccine that is also adjuvanted with IL7 and IL15? This could be discussed in a little more detail
  2. Fig 1G; The text mentions SIV Gag p27 values peaked on day 3, but nothing beyond day 3 is shown. Is there data to show values beyond day 3? This figure does not have statistical information. On day 3,  IL7 expressing virus seems to have more SIV Gag p27 expression- is this not significant? How about co-transfection of IL7 and 15 similar to fig. 2A
  3. Line 426-427 states no response after first dose- is this data similar to what was previously reported?
  4. ADCC should be spelled out
  5. Line 540-544 should include citations

Author Response

We thank this reviewer for finding the manuscript well written, and the experiments well thought with appropriate controls.

Comment 1:  What is the advantage/difference of this vaccine over the DermaVir, a HIV-1 Gag based vaccine that is also adjuvanted with IL7 and IL15? This could be discussed in a little more detail.

This is a pertinent question addressed by this reviewer. We now introduced in the text more information about the DermaVir using HIV DNA as vaccine. Briefly, DermaVir is a biomedicine patch that associates variable HIV DNA constructs as needle-free skin delivery to target the dermal population of antigen-presenting cells. One of these DNA constructs resembles our parental CAL-SHIV-IN- that is lacking integrase. However, this HIV DNA vaccine prototype was driven by the Tat-dependent SIV/HIV LTRs while our prototype is driven by the constitutive CAEV LTRs. The cytokine studies with DermaVir were not coexpressing both HIV antigens with the cytokines. In this present study, we associated this coexpression to favor the immune cell and cytokine interactions during the Ag-presentation in a immunologic synapse.

Comment 2: Fig 1G; The text mentions SIV Gag p27 values peaked on day 3, but nothing beyond day 3 is shown. Is there data to show values beyond day 3?

We apologize for using the wrong terminology on describing the increase of SIV Gag p27. This has been corrected in the manuscript, removing “peaked…” with “highest…”.

Comment 3: This figure does not have statistical information. On day 3, IL7 expressing virus seems to have more SIV Gag p27 expression- is this not significant?

We fully agree with this reviewer that this figure is lacking the statistical analysis. However, the n-value was not enough to run the statistical analysis.

Comment 4: How about co-transfection of IL7 and 15 similar to fig. 2A

This is an excellent suggestion; however, this analysis was not included in the experimental procedure.

Comment 5: Line 426-427 states no response after first dose- is this data similar to what was previously reported?

In our previous studies, ICS assay using the lentiDNA CAL-SHIV-IN- was performed only on humanized hu/NOD/SCID/b2 mice, but not on BALB/c that we used in this study. In the humanized mice, we could detect secretion of IFN-g by T cells as early as 1 to 2 weeks post-immunization.

Comment 6: ADCC should be spelled out

Thanks to this reviewer, we have now spelled out ADCC in the text.

Comment 7: Line 540-544 should include citations

This reviewer is right, we have now included citations in lines 540-544.

Reviewer 2 Report

Leroy et al. tested a novel vaccine using a lentivirus/DNA-based vaccine in which IL-7 and IL-15 DNA cassettes were added as a means to enhance T cell responses to immunization. This vaccine was tested in both mice and rhesus macaques, and compared to the parental vaccine which lacked the cytokine cassettes. The authors make a good case for this strategy in the introduction, and their experimental approach to compare the parental vaccine to the cytokine-adjuvanted vaccine in mice and NHP is sound. It was satisfying to see the testing of the parental LentiDNA and cytokine-adjuvanted version of the vaccine in in vitro studies (Figures 1 and 2). In a small study of mice (N=4-5 mice per group per time point), the cytokine-adjuvanted vaccine yielded a greater IFNg and IL-2 response than the parental vaccine at week 10. Humoral responses in mice are not shown.

In macaques, the effects of the cytokine adjuvants are less clear, especially considering there were only 3 animals per experimental group. In ELISpot analyses, there appears to be a spike in SIV-Gag specific T cells at week 26 (Figure 4A), but this difference in IFNg-producing cells between groups 1 and 2 is not detected in ICS assays of CD4 and CD8 T cells at the same time point(Fig. 4B) – this discrepancy in assay read-outs is not addressed, and dampens enthusiasm about the ELISpot result. On top of that, with only 3 animals per group, no indication of variability, and what appears to be very low T cell responses, it’s difficult to draw any conclusions from Figure 4b and c. The serological analyses do not reveal notable differences in humoral responses, with the possible exception of the IgG and IgA titers in rectal secretions (Figure 5e) , but with only 3 animals per group and no depiction of the biological variability across NHP, it’s difficult to get excited about these results.

The major strengths of this study are the scientific question and experimental approach – the authors have an interesting question about the use of cytokines as adjuvants, and use the appropriate animal models and assays to answer them. The major weakness, however, is that the animal studies are underpowered. With only 4-5 mice per group per time point (from one experiment) and 3 macaques per experimental group, the studies would, at best, reveal only qualitative differences in immune responses to the parental vs. cytokine-adjuvanted vaccine. However, there do not appear to be consistent, qualitative differences in responses to the parental vs. cytokine-adjuvanted vaccines, so enthusiasm about the addition of cytokines to the vaccine is low.

In the end, I had a hard time coming to a conclusion about what should happen next – with the low numbers of animals, inconsistencies in some experimental read-outs, and lack of obvious differences in immune responses between experimental groups, it was not obvious if the addition of cytokines to the vaccine offers any benefit.

Major Critiques:

  1. Animal studies are underpowered.
  2. In several figures (Figures 1g, 2c, 4b and c, 5e), the median values of assay read-outs for experimental groups are depicted without any indication of variability. It is difficult to make any conclusions about the data without any measure of experimental or biological variability. It seems that Figure 5e has the most potential to reveal differences between the parental and cytokine-adjuvanted vaccine in NHP (even with N of only 3), but without showing the variability in the data sets, it is difficult to be enthusiastic about the data.
  3. It appears that the cytokine-adjuvanted vaccine may have affected humoral responses more than T cell responses in NHP. Does the cytokine-adjuvanted vaccine impact humoral responses in mice?

Minor Critiques:

  1. For small sample sizes without measures of normality and variance, the unpaired t-test is unreliable for statistical comparisons. A non-parametric test would be more appropriate.
  2. There are instances where depiction of median data in figures appears to be inconsistent. For example, in Figure 3b, it appears that the correct median value is shown for IL-21 at week 10 for the Nef-specific CD4 T cell responses (where N=4), but the median is not shown correctly for the same time point for Nef-specific CD8 T cells responses (no bar is shown for the median of the 4 samples). Also, in Figures 5a and 5c, the legend claims that median values are shown, but it appears that mean (or possibly geometric mean) values are depicted.

Author Response

We thank this reviewer for finding that we have made a good case for the strategy in the introduction and the experimental approach to compare our parental and cytokine-adjuvanted vaccines. We also thank this reviewer for finding major strengths on scientific and experimental approach about cytokines as adjuvants to DNA vaccine.

Comment 1: Animal studies are underpowered

We fully understand the criticism of this reviewer with respect of the low numbers of animals used in this study. This is a pilot study in which our main aim was to evaluate the high potential increase of immunogenicity associated with the coexpression of cytokines with vaccine antigens. We fully agree with this reviewer that 5 mice per group are in the lower limit for powerful statistical analysis. However, due to the funding limits for this pilot study, we were not in position to increase further the number per group. Similar limitations apply for rhesus macaque experiments. We compensate the low number of animals by sequential longitudinal analyses

Comment 2: In several figures (Figures 1g, 2c, 4b and c, 5e), the median values of assay read-outs for experimental groups are depicted without any indication of variability. It is difficult to make any conclusions about the data without any measure of experimental or biological variability. It seems that Figure 5e has the most potential to reveal differences between the parental and cytokine-adjuvanted vaccine in NHP (even with N of only 3), but without showing the variability in the data sets, it is difficult to be enthusiastic about the data.

We are very thankful to this reviewer for this comment that helped us to revise the figures according to his suggestions. We now have included all the suggested modifications in the new version of the text.

Comment 3: It appears that the cytokine-adjuvanted vaccine may have affected humoral responses more than T cell responses in NHP. Does the cytokine-adjuvanted vaccine impact humoral responses in mice?

This is a very good question addressed by this reviewer. We did not observe positive effect of cytokines on the humoral response in mice. This is probably due to the fact that these are macaques cytokines inserted in the lentiDNA, which may have effects on macaque but not mouse B cells. In addition, the lentiDNA vaccine undergoes one cycle replication in macaques but not mice cells. This one cycle might produce more antigens that target more the B cells in macaques. We now have included all this information and the Ab responses in mice in this version of the manuscript.

Comment 4: For small sample sizes without measures of normality and variance, the unpaired t-test is unreliable for statistical comparisons. A non-parametric test would be more appropriate.

We are very thankful to this reviewer for this suggestion with respect of statistical analysis. We now have included non-parametric statistical tests in the new version of the manuscript.

Comment 5: There are instances where depiction of median data in figures appears to be inconsistent. For example, in Figure 3b, it appears that the correct median value is shown for IL-21 at week 10 for the Nef-specific CD4 T cell responses (where N=4), but the median is not shown correctly for the same time point for Nef-specific CD8 T cells responses (no bar is shown for the median of the 4 samples). Also, in Figures 5a and 5c, the legend claims that median values are shown, but it appears that mean (or possibly geometric mean) values are depicted.

Again, we thank this reviewer for this constructive comment. We have now corrected all discrepancies between medians and means, and we included the median bar on figure 3b, as suggested.